# U^2^-Net and ResNet50-Based Automatic Pipeline for Bacterial Colony Counting

**DOI:** 10.3390/microorganisms12010201

**Published:** 2024-01-18

**Authors:** Libo Cao, Liping Zeng, Yaoxuan Wang, Jiayi Cao, Ziyu Han, Yang Chen, Yuxi Wang, Guowei Zhong, Shanlei Qiao

**Affiliations:** 1Center for Global Health, Nanjing Medical University, Nanjing 211166, China13164067226@163.com (Y.W.);; 2Department of Pathogen Biology, School of Basic Medical Sciences, Nanjing Medical University, Nanjing 211166, China; zengliping.118@163.com

**Keywords:** bacterial colony counting, light intensity correction, convolutional neural networks, U^2^-Net, ResNet50, image segmentation, image spatial normalization

## Abstract

In this paper, an automatic colony counting system based on an improved image preprocessing algorithm and convolutional neural network (CNN)-assisted automatic counting method was developed. Firstly, we assembled an LED backlighting illumination platform as an image capturing system to obtain photographs of laboratory cultures. Consequently, a dataset was introduced consisting of 390 photos of agar plate cultures, which included 8 microorganisms. Secondly, we implemented a new algorithm for image preprocessing based on light intensity correction, which facilitated clearer differentiation between colony and media areas. Thirdly, a U^2^-Net was used to predict the probability distribution of the edge of the Petri dish in images to locate region of interest (ROI), and then threshold segmentation was applied to separate it. This U^2^-Net achieved an F1 score of 99.5% and a mean absolute error (MAE) of 0.0033 on the validation set. Then, another U^2^-Net was used to separate the colony region within the ROI. This U^2^-Net achieved an F1 score of 96.5% and an MAE of 0.005 on the validation set. After that, the colony area was segmented into multiple components containing single or adhesive colonies. Finally, the colony components (CC) were innovatively rotated and the image crops were resized as the input (with 14,921 image crops in the training set and 4281 image crops in the validation set) for the ResNet50 network to automatically count the number of colonies. Our method achieved an overall recovery of 97.82% for colony counting and exhibited excellent performance in adhesion classification. To the best of our knowledge, the proposed “light intensity correction-based image preprocessing→U^2^-Net segmentation for Petri dish edge→U^2^-Net segmentation for colony region→ResNet50-based counting” scheme represents a new attempt and demonstrates a high degree of automation and accuracy in recognizing and counting single-colony and multi-colony targets.

## 1. Introduction

Colony detection is an important and routine task in quality inspection for microbiologists, including food (e.g., microbial limit test), medicine, cosmetics, water, production equipment, air quality monitoring, and quarantine [1,2,3]. Currently, the majority of the laboratories are still using the method of pouring sample suspensions with agar into plates or streaking the agar plates with sample suspensions to culture colonies. However, the traditional visual check method for colony detection is laborious and time-consuming [4]. To reduce labor and improve analysis accuracy, researchers and developers have focused on image analysis methods. The main challenges in the methods are colony image acquisition, image segmentation, and classification of complex colonies. Furthermore, adjusting manual parameters is often necessary, yet it proves challenging for users aiming to enhance recognition accuracy.

Traditional algorithms, such as thresholding, watershed, and wavelet transform, have been widely used for segmenting the colony area in an image [5,6,7,8,9,10]. However, these algorithms often exhibit poor performance when dealing with images that have low contrast, noise, and/or adhesive colonies. In recent years, advancements in machine learning have yielded an attractive research field in the microbiology discipline [11,12,13]. Deep learning approaches, such as convolutional neural networks (CNNs), which mimic the information transmission mechanism of the visual system and learn features from training samples, are very good at image processing, especially in image classification, target retrieval, positioning detection, and target segmentation [14,15,16,17].

Over the years, research has increasingly focused on developing CNN techniques for counting bacterial colonies on agar plates [18]. For instance, two CNN approaches for bacterial colony counting were implemented, one based on a support vector machine (SVM), and the other utilizing a CNN architecture within the BVLC Caffe framework. These approaches achieved an impressive overall accuracy of 94.5% and demonstrated a notable improvement in counting multiple colony aggregates [19]. A modified U-Net architecture, incorporating a pre-trained ResNet34 network, was employed to quantify white/red colonies across 492 pairs of plate images, excluding areas with multiple small and overlapping colonies [20]. In addition, the Annotated Germs for Automated Recognition (AGAR) dataset, comprising over 330,000 labeled microbial colonies, was established [21]. The performances of Faster R-CNN [22] and Cascade R-CNN were then evaluated [23] on this dataset. Cascade R-CNN achieved the highest mean average precision (MAP) of 52.3% and 59.4% on intersection over union (IoU) ranging from 0.5 to 0.95. Furthermore, a lightweight improved YOLOv3 [24] network based on the few-shot learning strategy was proposed [25]. The network was trained and validated using only five raw images, resulting in an improvement in average accuracy from 64.3% to 97.4% and a significant decrease in the false negative rate from 32.1% to 1.5%.

Among the various neural network models, U^2^-Net [26] is a network structure based on U-Net [27] that is widely used for foreground recognition in medical images. The network architecture follows an encode–decode framework, incorporating elements from an FPN (feature pyramid network) [28] and U-Net. The authors introduced a novel module called RSU (ReSidual U-blocks), which has demonstrated excellent segmentation performance. Each RSU functions as a small U-Net, and they are interconnected in a structure similar to FPN, employing a top-down approach to enhance multi-scale capability. ResNet [29] is a CNN-based image classification architecture that effectively extracts feature information and exhibits faster convergence and higher accuracy. It is built on the concept of residual learning, enabling the network to become deeper and to achieve improved accuracy. In the official PyTorch code, ResNet offers five different depth options: 18, 34, 50, 101, and 152. The depth of each network refers to the number of layers that are trained and updated, including convolution layers and fully connected layers.

Building upon the achievements of prior research, our objective is to present a comprehensive and accessible approach for microbiologists, encompassing the entire process (Figure 1) from colony image acquisition to fully automated colony counting. The specific goals of this study are as follows: (i) designing and configuring hardware for imaging acquisition, (ii) culturing various bacterial species on agar plates and capturing photographs to create a dataset of bacterial colony images, (iii) employing the U^2^-Net for region of interest (ROI) [30] searching and colony region segmentation, and (iv) utilizing the ResNet50 framework for colony counting.

## 2. Materials and Methods

### 2.1. Bacterial Culture

The following bacterial strains were collected: Gram-negative bacteria *Escherichia coli* ATCC25922, *Salmonella typhimurium* ATCC14028, *Vibrio parahaemolyticus* ATCC17802, and *Shigella* sp. ATCC12038; Gram-positive bacteria *Staphylococcus aureus* ATCC6538, *Staphylococcus epidermidis* ATCC12228, *Listeria ivanovii* ATCC19119, and *Listeria monocytogenes* ATCC19115. Each bacterium was activated by streaking on plate count agar (PCA) or nutrient agar media in a 9 cm (diameter) Petri dish and the bacteria were cultured at 35–37 °C for 24–48 h to allow colony formation. Single colonies with good morphology were diluted using normal saline. Then, 100 μL aliquot of the bacterial suspension was inoculated on media by using the spread plate method to ensure the growth of approximately 0–300 colony forming units (CFU) on each plate. The bacteria were then incubated at 35–37 °C for 24–48 h. Strains, media, and Petri dishes were purchased from Huankai Microbial Sci. and Tech and the Guangdong Microbial Culture Collection Center (GDMCC), Guangzhou, China.

### 2.2. Assembly of the Imaging Device

As is known, when capturing images from the same plate using a reflected light camera, the image quality varies significantly due to different surfaces and lighting conditions. Therefore, it is necessary to standardize shooting conditions in the design of the image capture instrument. The performance parameters of the CMOS sensor and zoom lens (purchased from Shenzhen Shunhuali Electronics Co., Ltd., Shenzhen, China) (Figure 2B,C) are listed in Appendix A. An acrylic backlight panel (BLP) (Figure 2E), with LED lights, was used to ensure the brightness and uniform illumination of the Petri dish from the bottom. For the sake of mobility and portability, a Raspberry Pi (equipped with a 64-bit, 1.5 GHz, four-core, 28 nm CPU, purchased from Shenzhen Trxcom Electronics Co., Ltd., Shenzhen, China) was chosen (Figure 2A,D) for data acquisition, providing flexibility and adaptability to various experimental scenarios.

The distance between the lens and the Petri dish was determined based on the zoom range of the lens. Subsequently, a three-layer enclosed cabinet was designed and constructed using acrylic boards. The lower box functions as a black opaque structure to house the light sources. The middle box serves as a platform for positioning the Petri dish. The transparent upper box is designed to contain the CMOS camera, lens, Raspberry Pi, and wiring. The price of this device is less than USD 750.

### 2.3. Raw Image Acquisition

Bacterial cultures were positioned on the sampling table of the imaging device, and a Python program was utilized to capture a live image of the Petri dish. Once the light source and instrument had stabilized, the focus was adjusted manually or automatically to achieve optimal sharpness, and photos were taken using transmission spectral imaging. Subsequently, 100 frames were captured, and the average pixel value of these frames was calculated to reduce the signal-to-noise ratio. The captured raw image (Figure 3) size is 2560 × 1922 pixels.

### 2.4. Image Preprocessing

The color of an image can be influenced by data acquisition conditions, such as brightness, contrast, white balance, etc. Preprocessing images to a similar color can help improve the subsequent recognition accuracy of CNN models. Initially, the Petri dish area is approximately segmented from the image by steps including image denoising (range-based adaptive bilateral filter [31]), differential transformation, and threshold segmentation. Subsequently, through edge detection, the colony region and culture medium region were roughly separated. Considering that differences in the composition and thickness of the media between batches have an impact on the luminosity data that cannot be ignored, the average luminosity of all pixels in the culture medium region is represented by *I*_0_ and the luminosity of a pixel at specific location in the entire image is represented by *I_i_*. The corrected intensity data for each image were subsequently expressed as Formula (1). Figure 4A illustrates the flowchart of the data preprocessing, and the effect of preprocessing is shown in Figure 4B.
(1)$$lgI_{i^{\prime }}=lgI_{0}-lgI_{i}$$

### 2.5. Colony Identification and Counting

The process of colony identification and counting in this paper is mainly divided into three steps. In the first step, the edge of the Petri dish (foreground) is segmented using a U^2^-Net model. Along the inner edge of the extracted contour, the ROI (culture medium and colony region) is separated. The output of this U^2^-Net then undergoes threshold segmentation to obtain the foreground mask. Usually, pixels with gray values greater than the threshold are marked as the foreground (target), and pixels with gray values less than or equal to the threshold are marked as the background (non-target). In the second step, the colony region (foreground) is segmented from culture medium region (background) using another U^2^-Net model, following the same masking approach. Afterward, each connected region obtained by connected component labeling is regarded as a colony component (CC), which may contain a single colony or adhesive colonies. These CCs are then rotated, and the corresponding image crops are standardized to a size of 128 × 128 pixels. In the third step, these standardized image crops are inputted to ResNet50, which extracts features and outputs the number of colonies. The workflow is briefly illustrated in Figure 1.

#### 2.5.1. Culture Dish Edge Segmentation

##### Dataset Preparation and Network Construction

To train U^2^-Net for extracting the edges of the Petri dishes, we manually annotated the foreground regions of the preprocessed images: the inner and outer edges of Petri dishes were labeled, and the pixel values between the 2 circles were filled with a pixel value of 255, resembling Figure 5Ab. A total of 255 colony images were annotated, with 219 images used as the training set and 36 images used as the validation set.

To improve the training efficiency of the model, the RGB values were normalized to a range between 0 and 1. Since the light intensity values were already adjusted to a recognizable distribution, the typical standardization operation based on mean and standard deviation was not applied here.

##### Network Training

The U^2^-Net model we used is initialized with random parameters. AdamW was employed as an optimizer with a learning rate of 1 × 10^−3^, betas (0.9, 0.999), and epsilon (eps) of 1 × 10^−8^. The learning rate decay strategy employed for the model is cosine learning rate decay [32]. The chosen loss function was binary cross Entropy with logits loss (BCEWithLogitsLoss) [33]. To prevent overfitting, a regularization term with a weight decay of 1 × 10^−4^ is incorporated to the loss function. The batch size is set to 1.

To address the class imbalance between the foreground and background, we increased the weight of the foreground in the loss function to enhance the accuracy of foreground recognition. Since the edge region of the Petri dish in the images occupies approximately 1/8 of the background area, we set a foreground-to-background weight ratio of 8:1 in the loss function.

To evaluate the model’s prediction performance on the validation set, we utilized mean absolute error (MAE) and F1 score as evaluation metrics [34]. The F1 score is the harmonic mean of precision and recall, where predicted values greater than 0.5 are considered as 1 and values less than 0.5 are considered as 0. The F1 score ranges between 0 and 1, with a higher value indicating better classifier performance. MAE is used to measure the magnitude of differences between the model’s predicted values and the ground truth. A smaller MAE indicates less discrepancy between the model’s predictions and the actual values. The calculations used are shown in Formulas (2)–(4) (*TP*: true positives; *FN*: false negatives; and *FP*: false positives). Formulas (2)–(4) are as follows:(2)$$Recall=\frac{TP}{TP+FN}$$
(3)$$Precision=\frac{TP}{TP+FP}$$
(4)$$F1=\frac{1.3*Precision*Recall}{0.3*Precision+Recall}$$

The output of U^2^-Net is the probability of each pixel being classified as foreground, and a threshold segmentation is performed to determine the final mask of foreground (the edge of the dish, Figure 5Ab). Subsequently, the ROI was separated by utilizing the generated mask (Figure 5Ac).

#### 2.5.2. Colony Region Separation

Through the previous process, we identified the ROIs (Figure 6Aa). However, these ROIs usually consist of colonies, culture media, and other impurities, such as stains or damaged culture media. To address this, we trained another U^2^-Net network for pixel-level recognition, with colonies as the foreground. Thereafter, we applied a threshold segmentation technique to the foreground probability map generated by this U^2^-Net [35], enabling us to precisely define the colonies (Figure 6Ab).

##### Dataset Preparation

We manually marked the pixels belonging to the colonies within the ROIs of 255 colony images, creating masks resembling those in Figure 6Ab. These masks served as labels to construct the dataset, with 219 images allocated for training and 36 images designated for validation.

##### Network Training

The images in the training set were horizontally flipped to augment the data (with a flip rate of 0.5). The RGB channel values of the images were scaled to the range of 0–1 using the same method as described above. The training parameters remained consistent with those used for the Petri dish edge segmentation process. After colony region extraction, we conducted threshold segmentation (Figure 7) to determine the final mask of the colony region.

#### 2.5.3. Colony Counting

##### Dataset Preparation

Using connected component analysis, we segmented the foreground (colony region) images obtained in the previous step into crops containing no colonies, single colonies, or adhered colonies (colony components) and manually annotated the number of colonies in each crop. The labels in the training and validation datasets included ten categories ranging from 0 to 9. If the number of colonies was 9 or more, it was uniformly labeled as 9. Crops devoid of colonies were assigned a label of 0. Among the labels, 76.6% contained isolated colonies, 9% contained 2 colonies, 2.2% contained 3 colonies, 1.2% contained more than 3 colonies, and 10.9% contained 0 colonies.

To improve the recognition accuracy of ResNet50, spatial normalization was applied to the CCs (Figure 8A). By fitting an ellipse around the CC and taking the center of the ellipse as the rotation center point O, the CC was rotated by an angle θ to align the major axis of the ellipse vertically. This rotation operation ensured consistent spatial characteristics, making the distribution of adhesive colonies more uniform.

For training ResNet50, the dimensions of image crops were standardized to 128 × 128 pixels. Images smaller than 128 × 128 pixels were padded with zeros to achieve the standard size. Images larger than 128 × 128 pixels were resized by reducing the longer side to 128 pixels while maintaining the aspect ratio. The shorter side was then padded with zero to reach a length of 128 pixels. This approach preserved the colony features while ensuring that all image crops were a consistent size. In total, 14,921 image crops were used for training, and 4281 image crops were used for validation.

##### Network Architecture

Given variations in runtime and classification accuracy across different ResNet versions, the well-used and robust ResNet50 was selected. To accommodate the prediction of 10 categories (class 0–9), we modified the output channels of the final layer to 10.

##### Network Training

Similar to the previous two processes, the RGB channel values of all images were scaled to the range of 0–1. The optimizer used for training is Adam, with a learning rate of 0.0001. The chosen loss function was BCEWithLogitsLoss. To improve accuracy, we also introduced different weights for each annotated category when computing the loss to balance the sample quantities among categories. The weight for each class was calculated using the following Formula (5) (where *w_i_* is the weight for annotated category *®*, and *n_i_* is the sample quantity for annotated category *i*):(5)$$w_{i}=\frac{max\left(n\right)}{n_{i}}$$

### 2.6. Training Environment

This study was conducted using PyCharm and Python 3.9. The training and testing of the models were performed on a deep learning workstation equipped with an Intel^®^) X^®^ (R) CPU E5-2680 v4 processor and a NVIDIA Tesla P100 GPU with 16 GB of memory. The workstation was running Windows 10, and GPU acceleration was achieved using CUDA 10.0 and cuDNN 7.6. We implemented our models using Torch 1.12.1 and Torchvision 0.13.1.

## 3. Results

### 3.1. The Imaging System Captures High-Quality Colony Images

The high-resolution photographs used in this study were captured using the imaging device, which comprises a 5.1 Mpx CMOS sensor camera with a 2.8–12 mm lens and LED source light mounted below the Petri dish (Figure 2). The darkroom structure eliminates the influence of ambient light and enhances illumination uniformity. Compared to similar workstations, it has a relatively affordable price and is highly portable for various remote application scenarios.

We successfully cultured eight common foodborne pathogens. Using this setup, we collected a total of 390 high-resolution raw RGB images of agar plates with a resolution of 2560 × 1922 pixels (Figure 3). By employing the transmission light source mode, reflections and shadows that may occur when using a mobile phone or camera for direct shooting were eliminated, emphasizing the ROI and providing sufficient contrast between the colonies and the background in the images. The distribution of original pictures of the eight bacterial species is as follows: 64 images for *S. aureus*, 66 images for *E. coli*, 38 images for *L. monocytogenes*, 55 images for *S. typhimurium*, 32 images for *L. ivanovii*, 60 images for *Shigella* sp., 48 images for *S. epidermidis*, 20 images for *V. parahaemolyticus*, and 7 images for colony cultures of environmental samples.

### 3.2. Image Preprocessing

Variations in the composition and thickness of the media can profoundly influence image luminosity, contrast, and colony recognition factors that are frequently underestimated. Following the capture of original images, a subsequent preprocessing step (Figure 4A) was applied. Subsequently, the correction of light intensity data for each image, using Formula (1), was employed to mitigate interference from the media and Petri dish. This correction enhanced the contrast between colonies and the background, revealing more intricate colony features (Figure 4B), which were then extracted for CNN training.

### 3.3. Petri Dish Edge Segmentation

Considering the existing challenges in colony analysis, such as colony size, adhesion, and potential residual noise after preprocessing, neural networks were employed in the subsequent steps. We first utilized U^2^-Net to extract the edges of the Petri dish (Figure 5Aa,b) and located the central region, referred to as the ROI (Figure 5Ac). The model’s prediction performance on the validation set was evaluated by the MAE and F1 score.

The F1 score and MAE curves of the model on the validation set for each epoch are shown in Figure 5B. The model achieves a maximum F1 score of 99.5% and a minimum MAE of 0.0033, indicating extremely accurate foreground predictions. The parameters that achieved the highest F1 scores and the lowest MAE were chosen as the optimal parameters.

Different threshold values can influence the size of the ROI. To determine the threshold that best suits the requirements, we performed threshold segmentation tests on the output of U^2^-Net using thresholds ranging from 0.1 to 0.9, as well as 0.95 and 0.99. Boxplots of MAE, precision, and recall illustrate the distribution of segmentation results for different thresholds. As shown in Figure 5C, as the threshold increases, MAE reaches its minimum value at 0.7, precision gradually increases, and recall gradually decreases. These results indicate that lower thresholds yield more complete and reliable ROIs [36]. A relatively lenient delineation of the edges implies fewer residuals, preventing U^2^-Net from mistakenly identifying edges as colonies and thereby affecting colony counting. Lower threshold settings also improve generalization while maintaining MAE within an acceptable range. Therefore, we selected 0.1 as threshold in this study.

### 3.4. Colony Region Separation

After obtaining the ROI (Figure 6Aa,b), we used another U^2^-Net model to extract the colony regions within the ROI. The curve in Figure 6B represents the F1 score and MAE on the validation set for each epoch. This U^2^-Net model achieved a maximum F1 score of 96.5% and a minimum MAE of 0.005 on the validation set, indicating its accurate extraction of colonies.

Regarding the threshold selection for postprocessing of extracted colony region, we also conducted segmentation tests using multiple thresholds on the validation set. As shown by the red arrow in Figure 7A, setting the threshold to 0.2 resulted in false-positive colonies, primarily located at the edges of the ROI. These false-positive results mainly represented edges of the medium region. As the threshold increased from 0.2 to 0.999, the number of false-positive colonies significantly decreased, and higher thresholds improved the ability to segment multiple adhesive colonies (as indicated by the white arrow in Figure 7A), thereby reducing the burden and errors of ResNet50. However, setting the threshold too high (0.99 and 0.999) resulted in some colony regions being partly classified as background, as indicated by the blue arrow in Figure 7A, leading to an increase in MAE. The boxplots of MAE, precision, and recall illustrate the distribution of segmentation results at different thresholds, as shown in Figure 7B. With the increase in threshold, MAE, precision, and recall, respectively, exhibited the same trend as in Figure 5C. In the task of colony region extraction, compared to Petri dish edge segmentation, it is more crucial to achieve low MAE and high precision predictions in order to accurately segment colonies and minimize the false-positive rate. Taking all factors into consideration, we selected 0.9 as the threshold, which provides a low false-positive rate for the majority of colony images and accurately segments adhesive colonies.

### 3.5. Training and Validation Performance of ResNet50 on Colony Counting

The spatial normalization used in this study (Figure 8A) helps to enhance the spatial consistency of CCs, alleviate the counting pressure on ResNet50, and improve the model’s recognition accuracy. When half of the data were not rotated, ResNet50’s performance was slightly inferior. When all data were randomly rotated by 45 degrees, ResNet50 exhibited the poorest performance. In comparison, when all CCs were subjected to our spatial normalization, ResNet50 achieved faster training speed, a lower loss curve, and higher accuracy on the validation set (Figure 8B,C).

### 3.6. Test Performance of the Pipeline on Colony Counting of Entire Images

This study used manually annotated results as the ground truth and the predicted colony numbers as the predicted values to construct the confusion matrix for the entire counting process on the test set (Table 1). The recovery calculated for class > 0 was shown in Equation (6), where *n_true_* and *n_pred_*, respectively, represent the actual and predicted bacterial counts. For the class = 0 scenario, given the absence of colonies in the image crops, the recovery evaluation is treated as a binary classification. Here, *TP* represents the number of image crops correctly predicted as class 0, and *FN* represents the number of image crops predicted as other classes.
(6)$$Recovery=\left\{\begin{array}{c} 1-\frac{\left|n_{true}-n_{pred}\right|}{n_{true}},\;\;n_{true}>0\\ \;\frac{TP}{TP+FN},\;\;n_{true}=0 \end{array}\right.$$

Our approach achieved an overall recovery of 97.82%, with 66.55% of colonies segmented into single colonies and 10.21% of colonies segmented into two-adhesion colonies. The identification effect on image crops with no colonies and with 1–3 colonies was the best (recall: 83.62–97.59%; precision: 89.80–98.88%; recovery: 94.54–99.90%), but the identification effect on adhesions with more than three colonies began to decline. However, most of the incorrect predictions are close to the actual values. Additionally, counting these adhesive colonies is often challenging even for experienced technicians.

Figure 9 illustrates the results of the entire automated counting process depicted in Figure 1. Each white box represents a CC, and the number displayed on the box represents the colony count, demonstrating the ability to accurately identify isolated colonies and perform well on most adhesive colonies. Additionally, the hyperparameters for the entire colony counting process have been fine-tuned for various scenarios, eliminating the need for further adjustments.

## 4. Discussion

### 4.1. U^2^-Net Is More Suitable for Locating the Edges of Petri Dishes and Extracting Bacterial Colony Areas Compared to Threshold Segmentation

U^2^-Net has been successfully applied in complex biomedical images [37] and is utilized to generate the density map for counting microbiological objects in images [38]. In this study, the task of U^2^-Net is to identify the Petri dish edges and then extract colony regions in images, while traditional algorithms, such as k-means [39] and Otsu’s binarization [40], typically employ thresholding methods. We compared the impact of U^2^-Net with that of k-means and Otsu’s image processing methods on the colony counting performance of ResNet50. Consequently, ResNet50 achieved recovery of 99.29%, 87.87%, and 42.98% on the validation dataset, respectively. It is evident that images processed with U^2^-Net yielded better colony counting results. Furthermore, when compared to the results obtained using the traditional watershed algorithm and a proposed CNN algorithm [19], ResNet50 demonstrated superior precision and recall in counting aggregated colonies (Appendix A).

### 4.2. The ResNet50 Model in Our Proposed Method Functions as an Interchangeable Module

In our proposed counting process, ResNet50 is tasked with colony counting. We conducted training and validation on ResNet50, ResNet_wide [41], VGG19 [42], and EfficientNet [43], employing identical parameters. After 200 epochs of training, the accuracy for each model is as follows: 96.24% for ResNet50, 96.35% for ResNet_wide, 96.05% for VGG19, and 96.45% for EfficientNet, suggesting negligible distinctions. Consequently, the ResNet50 module is an interchangeable constituent in our colony counting methodology.

### 4.3. Our Method Surpasses YOLO, the Segment Anything Model, and OpenCFU in Performance

YOLO is a single-stage object detection architecture known for its simplicity and speed. With *S. aureus* images from the AGAR dataset, YOLOv5 achieved a mAP@0.5 of 99.1% [44]. We trained an official open-source YOLOv5 model (https://github.com/ultralytics/yolov5/tree/master, accessed on 22 November 2022) and evaluated its performance using the same datasets that ResNet50 used. The performance of colony counting was assessed based on three metrics: recovery, the number of false-positive colonies, and the number of false-negative colonies. On the validation set (36 images with a total of 1410 colonies), our model counted a total of 1420 colonies with a recovery of 99.29%, 20 false-positive colonies, and 19 false-negative colonies. In contrast, YOLOv5 counted a total of 1394 colonies with a recovery of 98.87%, 48 false-positive colonies, and 120 false-negative colonies. Additionally, using adaptive thresholds and a target maximum diameter, with a minimum diameter set at 15 pixels, OpenCFU 3.8-BETA [45] counted a total of 1116 colonies with a recovery of 79.29%. In summary, in terms of colony counting, our method outperforms the YOLOv5 model and OpenCFU.

Currently, artificial intelligence technology is advancing rapidly, with the emergence of advanced segmentation networks, such as the Segment Anything Model (SAM) [46,47]. For a raw colony image (Appendix A), the SAM model with default parameters fails to achieve fine colony segmentation (Appendix A). By adjusting the parameters, the SAM model can relative accurately segment the colonies (Appendix A); however, this also leads to an increase in false positive colonies and a 13-fold increase in processing time. In comparison, we obtain more accurate results on complex colonies (Appendix A) with only 1/10th of the time required, demonstrating the reliability of the automated counting process employed in this study.

### 4.4. The Impact of Bacterial Colony Quantity and Size on the Performance of Our Method

The quantity and size of bacterial colonies in images may affect the performance of colony counting. Evaluation of the validation dataset revealed that an increase in colony aggregation tends to correlate with an increased MAE (MAE < 0.025) for U^2^-Net to identify and extract colony regions (Appendix A). Furthermore, an evaluation of the recovery metric for the entire colony counting process indicates that an increase in colony aggregation does not significantly compromise the performance of colony enumeration (Appendix A).

In Figure 3, the plates containing eight bacterial species can be categorized into Class A (*E. coli*, *S. typhimurium*, *Shigella*, *V. parahaemolyticus*) and Class B (*L. ivanovii*, *L. monocytogenes*, *S. aureus*, *S. epidermidis*) based on their morphological features. We assessed the performance of colony counting for these two bacterial classes in the test set using recovery as the evaluation metric. The average recovery for the images in Class A was determined to be 98.85%, while for Class B, it was 98.06%. Additionally, Appendix A presents the recovery of our method for images of eight bacterial species. These findings suggest that the colony size has a minor impact on colony counting in this context.

## 5. Conclusions

In this study, through improvements in hardware design and setup, the refinement of image preprocessing and dataset preparation strategies, and the training of CNN models, we present microbiologists with an accessible and reliable solution for automatic colony counting in the laboratory. The proposed novel automatic pipeline for bacterial colony counting, mainly consisting of “light intensity correction-based image preprocessing→U^2^-Net segmentation for Petri dish edge→U^2^-Net segmentation for colony region→ResNet50-based counting” demonstrates excellent performance (an overall recovery of 97.82% for colony counting) and is capable of effectively separating the majority of complex colonies into individual entities.

## Figures and Tables

**Figure 1 microorganisms-12-00201-f001:**
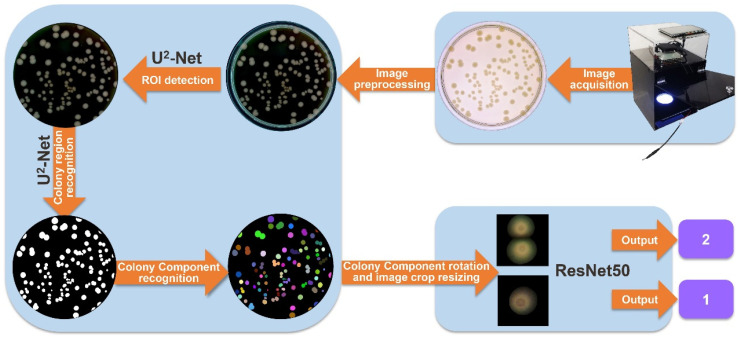
Overview of the processing workflow of the automatic pipeline for bacterial colony segmentation and counting in this study. The process begins with capturing a digital photo using the imaging system and preprocessing the image as input. The preprocessed image is then fed into U^2^-Net for ROI recognition and separation of the colony region from the media. Connected component analysis and cutting of the colony region result in image crops containing individual colony components (CCs). Finally, the CCs are rotated, and the image crops are resized as input for ResNet50 to perform the counting and obtain the colony numbers.

**Figure 2 microorganisms-12-00201-f002:**
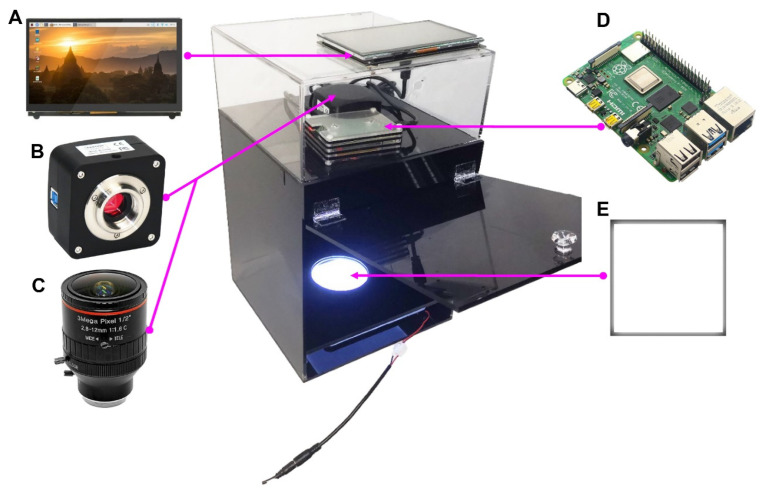
Actual photograph of the assembled colony imaging system. (**A**,**D**), Raspberry Pi; (**B**,**C**), CMOS camera and lens; (**E**) LED light source.

**Figure 3 microorganisms-12-00201-f003:**
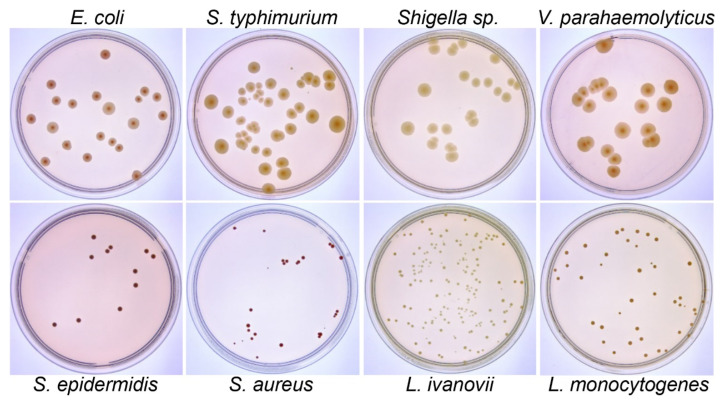
Example raw colony photos of eight bacterial species on agar plates taken by the image acquisition setup.

**Figure 4 microorganisms-12-00201-f004:**
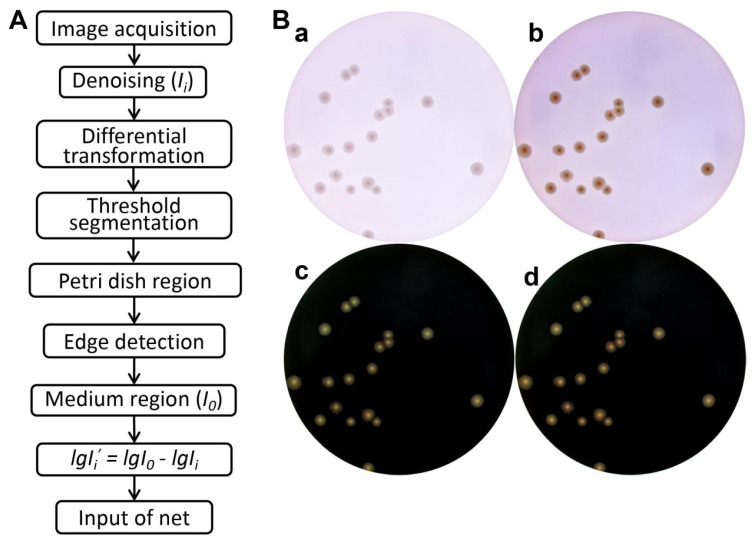
Image preprocessing workflow and its impact. (**A**) A flow diagram of the image preprocessing; (**B**) comparison of performance between two photos obtained at different times from the same plate before (**a**,**b**) and after (**c**,**d**) preprocessing.

**Figure 5 microorganisms-12-00201-f005:**
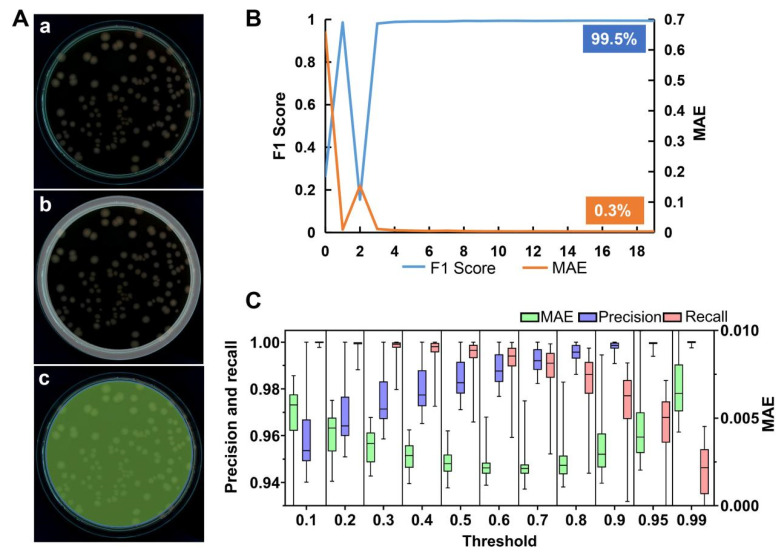
Overview of Petri dish edge segmentation results. (**A**) An example of the input image after preprocessing (**a**), highlighted Petri dish edge (**b**), and the central region of the Petri dish/ROI (**c**); (**B**) F1 score and MAE curve on the validation set; (**C**) a box plot comparing the Petri dish edge segmentation performance of U^2^-Net at different thresholds. *n* = 255. MAE, precision, and recall metrics were calculated separately for U^2^-Net at different thresholds. In the box plot, the center line represents the median, the bottom and top hinges represent the first and third quartiles, and the whiskers show the most extreme points within 1.5 times the interquartile range.

**Figure 6 microorganisms-12-00201-f006:**
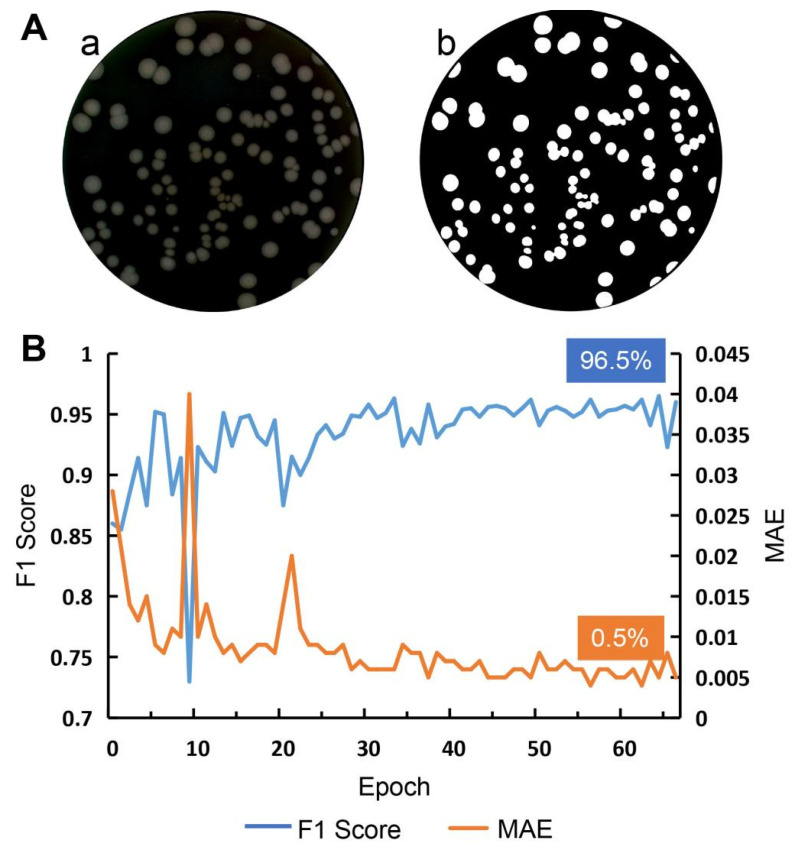
U^2^-Net performance for colony segmentation. (**A**) An example of the ROI after Petri dish edge segmentation (**a**) and the predicted result of the U^2^-Net model after threshold segmentation (**b**); (**B**) F1 score and MAE curve on the validation set.

**Figure 7 microorganisms-12-00201-f007:**
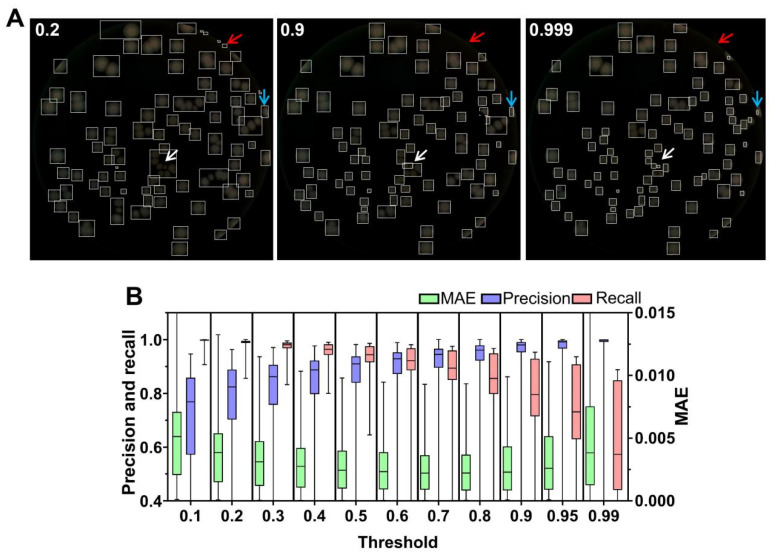
Comparison of colony segmentation performance at different thresholds. (**A**) An example of the results of connected component analysis after colony segmentation by U^2^-Net and threshold segmentation at different thresholds (0.2, 0.9, and 0.999). A threshold of 0.2 leads to incorrectly segmented edge objects (indicated by red arrows). With a threshold of 0.999, adhesive colonies are segmented more accurately (indicated by white arrows). However, increasing the threshold results in incorrect segmentation of colonies (indicated by blue arrows); (**B**) a box plot showing the segmentation performance comparison of U^2^-Net at different thresholds. *n* = 255. MAE, precision, and recall metrics were calculated separately for U^2^-Net at different thresholds. The median, bottom and top hinges, and whiskers carry the same meaning as described above.

**Figure 8 microorganisms-12-00201-f008:**
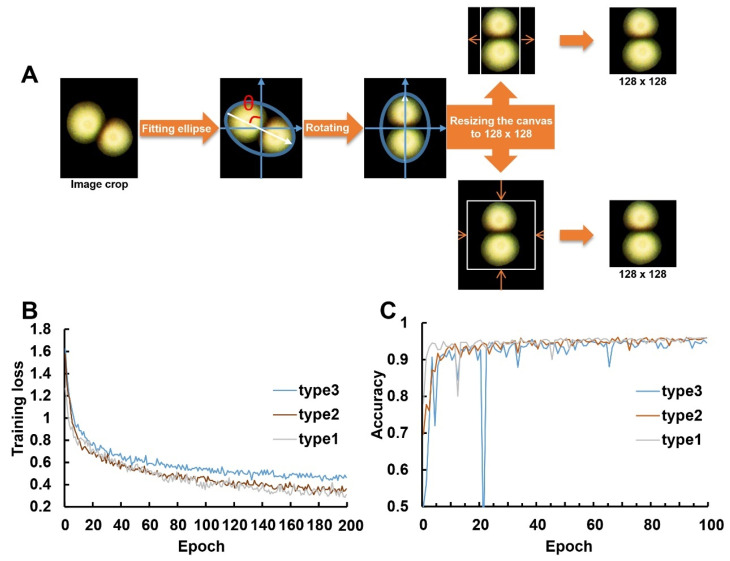
Overview of the CC rotation method and performance comparison of ResNet50 on image under different rotation treatments. (**A**) Schematic of the CC (indicated by the blue circle) rotation method; (**B**) loss curve for type1 (all data were rotated), type2 (half of the data were rotated) and type3 (all data were randomly rotated by 45 degrees) on the training set; (**C**) accuracy curve on validation set.

**Figure 9 microorganisms-12-00201-f009:**
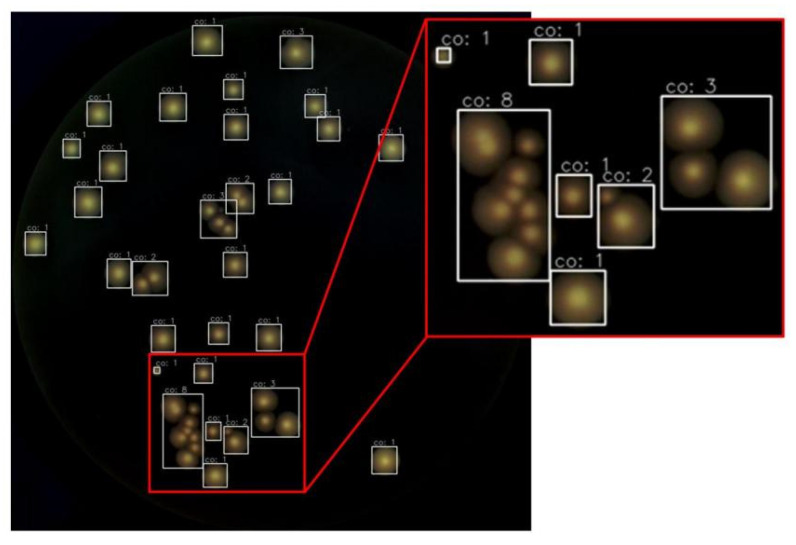
A result after applying the proposed method, showing improved recognition accuracy on complex colonies. The inset shows the enlarged view of the counting result for adhesive colonies.

**Table 1 microorganisms-12-00201-t001:** The confusion matrix, recall, precision, and recovery of the proposed method on test set.

Predict Class	Ground Truth									
	0	1	2	3	4	5	6	7	8	9
0	485	36	0	0	3	0	0	0	0	0
1	19	3002	10	3	1	0	1	0	0	0
2	0	37	458	15	0	0	0	0	0	0
3	0	1	4	97	0	0	0	0	0	0
4	0	0	0	0	58	6	0	0	0	0
5	0	0	0	1	3	56	5	0	0	0
6	0	0	0	0	0	0	71	2	1	0
7	0	0	0	0	0	0	4	103	1	0
8	0	0	0	0	0	0	0	0	61	3
9	0	0	0	0	0	0	0	0	0	75
Recall	96.23%	97.59%	97.03%	83.62%	89.23%	90.32%	87.65%	98.10%	96.83%	96.15%
Precision	92.56%	98.88%	89.80%	95.10%	90.63%	86.15%	95.95%	95.37%	95.31%	99.99%
Recovery	96.23%	99.90%	99.36%	94.54%	95.38%	98.06%	98.77%	99.73%	99.40%	99.57%

## Data Availability

The datasets used in this paper have been uploaded at: https://www.kaggle.com/datasets/clb2256095392/automatic-colony-counting/versions/1 (accessed on 18 July 2023). The algorithms we used can be download from: https://github.com/daimaku1/automatical-colony-counting/tree/master (accessed on 16 May 2023). The CNN-related code used in the paper is open source and appropriately cited, while the remaining code was developed by our team. This study received ethical approval from the Nanjing Medical University Biosafety Laboratory Certificate No. NJ2023184.

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
