# Peer review of "U2-Net and ResNet50-Based Automatic Pipeline for Bacterial Colony Counting"

_microorganisms, 2024, doi:10.3390/microorganisms12010201_

Round 1

Reviewer 1 Report (Previous Reviewer 1)

Comments and Suggestions for Authors

This paper presents an automated colony counting system utilizing an improved image preprocessing algorithm and a Convolutional Neural Network (CNN). The study involves the creation of an LED backlighting platform for image capture, generating a dataset of 390 agar plate culture photos with eight microorganisms. The proposed algorithm enhances image preprocessing through light intensity correction, achieving clearer separation between colonies and media areas. A U2-Net predicts the Petri dish edge to locate the region of interest, achieving an F1 score of 99.5%. Another U2-Net separates the colony region, achieving an F1 score of 96.5%. The colony area is then segmented into components for ResNet50-based automatic counting, yielding an overall recovery of 97.82%. This novel approach demonstrates high automation and accuracy in recognizing and counting single and multi-colony targets.

The article may be accepted in its current form.

Author Response

Thanks for reviewer’s time and efforts for giving us comments.

Reviewer 2 Report (Previous Reviewer 2)

Comments and Suggestions for Authors

The paper introduces  approach, which involves image preprocessing based on light intensity correction, U2-Net segmentation for the Petri dish edge, U2-Net segmentation for colony regions, and ResNet50-based counting. This comprehensive scheme represents a pioneering effort and showcases a high level of automation and accuracy in the recognition and quantification of both single and multi-colony targets.

The presented work requires significant revision, specifically:

1  In Figure 1,  the first image was obviously different from others, why not use the same image for illustrating,

2. In section 2.5.3.1, Dataset preparation, why here 10.9% contained 0 colonies ROI, and while Network Trainning get F1 score of more than 99.5% and MAE of 0.0033. Let readers confuse. Please classify.

3 For each microbial colony, the accuracy should be listed in table; and the analysis about the different results of big colony and small colony should be made.

4. The article's focus on deep learning for counting the number of bacterial colony, and consider the U2-net is suitable for count. But the result of the tranditional threshold method for counting should be  listed to proven this U2-net deep learning is suiable.

5. The absence of an ethics committee protocol for data acquisition is a concern.

As it stands, the work must be made a major revision, or it cannot be recommended for publication.

Author Response

The paper introduces approach, which involves image preprocessing based on light intensity correction, U2-Net segmentation for the Petri dish edge, U2-Net segmentation for colony regions, and ResNet50-based counting. This comprehensive scheme represents a pioneering effort and showcases a high level of automation and accuracy in the recognition and quantification of both single and multi-colony targets. The presented work requires significant revision, specifically: 

Response: Thanks for reviewer’s time and efforts for giving us comments.

Comment 1: 1. In Figure 1, the first image was obviously different from others, why not use the same image for illustrating

Response 1: The first image in Figure 1 was replaced with one identical to the others for illustrating.

Comment 2: 2. In section 2.5.3.1, Dataset preparation, why here 10.9% contained 0 colonies ROI, and while Network Trainning get F1 score of more than 99.5% and MAE of 0.0033. Let readers confuse. Please classify.

Response 2: The F1 score reflects the proportion of the area of misidentification relative to the entire ROI area. These misidentification areas consist of small residual Petri dish edges and impurities. In section 2.5.3.1, image crops containing such small residual Petri dish edges and impurities (as shown in Figure R1) account for 10.9% of the total. This percentage is derived from the area proportion of the misidentified regions relative to the entire colony area within the ROI, and this ratio is inversely related to the number of colonies in the ROI. These small residual Petri dish edges and impurities differ significantly from bacterial features and are easily classified by ResNet50 as non-colonies or 0 colonies.

Figure R1 Image crops containing small residual Petri dish edges and impurities.

Comment 3: 3 For each microbial colony, the accuracy should be listed in table; and the analysis about the different results of big colony and small colony should be made.

Response 3: In Figure 3, the plates containing 8 bacterial species can be categorized into Class A (E. coli, S. typhimurium, Shigella, V. parahaemolyticus) and Class B (L. ivanovii, L. monocytogenes, S. aureus, S. epidermidis) based on their morphological features. We assessed the performance of colony counting for these two bacterial classes in the test set using recovery as the evaluation metric. The average recovery for the images in Class A was determined to be 98.85%, while for Class B, it was 98.06%. These findings suggest that the morphology of the colonies has a minor impact on colony counting in this context. The counting recovery of our method for images of eight bacterial species is presented in Table S3. The analysis about the different results of big colony and small colony was in section 4.4 (Line438-Line455).

Comment 4: 4. The article's focus on deep learning for counting the number of bacterial colony, and consider the U2-net is suitable for count. But the result of the tranditional threshold method for counting should be listed to proven this U2-net deep learning is suiable.

Response 4: In this study, the task of U2-Net is to identify the Petri dish edges and extract colony regions in ROIs, while traditional algorithms such as k-means and Otsu's binarization typically employ thresholding methods. We compared the impact of U2-Net, k-means, and Otsu's-based image processing methods on the colony counting performance of ResNet50. Subsequently, ResNet50 achieved recovery of 99.29%, 87.87%, and 42.98% on the validation dataset, respectively. It is evident that images processed with U2-Net yielded better colony counting results (Section 4.1, Line393-Line403). Furthermore, when compared to the results obtained using the traditional watershed algorithm and a proposed CNN algorithm[1], ResNet50 demonstrated superior precision and recall in counting aggregated colonies (Table R1).

Table R1. Comparison of Colony Counting between ResNet50, Watershed, and a reference CNN Algorithms.

Class

A proposed CNN

Watershed

ResNet50

Precision

Recall

Precision

Recall

Precision

Recall

2

0.93

0.92

0.67

0.57

0.90

0.97

3

0.83

0.88

0.51

0.48

0.95

0.84

4

0.77

0.70

0.37

0.37

0.91

0.89

5

0.59

0.44

0.24

0.26

0.86

0.90

6

0.71

0.73

0.21

0.45

0.96

0.88

Comment 5: 5. The absence of an ethics committee protocol for data acquisition is a concern. As it stands, the work must be made a major revision, or it cannot be recommended for publication.

Response 5: Data acquisition in this study received ethical approval from the Nanjing Medical University Biosafety Laboratory Certificate No. NJ2023184. Data Availability Statements are in Lines 468 to 474.

  1. Ferrari, A.; Lombardi, S.; Signoroni, A. Bacterial colony counting with Convolutional Neural Networks in Digital Microbiology Imaging. Pattern Recognition 2017, 61, 629-640, doi:https://doi.org/10.1016/j.patcog.2016.07.016.

Round 2

Reviewer 2 Report (Previous Reviewer 2)

Comments and Suggestions for Authors

Please let the table R1 put in the manuscript as a comparison method to support the advantages of U2-net, and an appropriate literal statement should be included to use the traditional threshold or machine learning method.

Author Response

Comment : Please let the table R1 put in the manuscript as a comparison method to support the advantages of U2-net, and an appropriate literal statement should be included to use the traditional threshold or machine learning method.

Response: Thanks for the comments. Table R1 with appropriate literal statement was included in the new manuscript.

This manuscript is a resubmission of an earlier submission. The following is a list of the peer review reports and author responses from that submission.

Round 1

Reviewer 1 Report

Comments and Suggestions for Authors

The paper introduces  approach, which involves image preprocessing based on light intensity correction, U2-Net segmentation for the Petri dish edge, U2-Net segmentation for colony regions, and ResNet50-based counting. This comprehensive scheme represents a pioneering effort and showcases a high level of automation and accuracy in the recognition and quantification of both single and multi-colony targets.

The paper outlines the development of an innovative system capable of automatically counting bacterial colonies from images of agar plates. This advancement holds substantial promise for various scientific and laboratory applications due to its exceptional accuracy and automation capabilities.

The presented work requires significant revision, specifically:

1. A small dataset and validation set are used, with 4 missing classes, yet the conclusion indicates high performance. Table 1 reveals a significant class imbalance, with class 1 having values significantly greater than other classes, and instances of zero classes. Consequently, the conclusions and proposed approach are inaccurate. Addressing class imbalance and increasing the sample size by 2-3 times is imperative.

2. Image quality enhancement is essential, and images with double numbering of panels (e.g., Figures 1, 2, etc.) need adjustment. Separating them for improved clarity may be beneficial. Separate drawings for panels A and B are required, all drawings must be legible.

3. Results and discussion must remain distinct sections and should not be merged. The conclusion should provide a vision for research development.

4. The article's focus on machine learning for bacterial colony analysis necessitates comparison with existing studies in the field, especially since numerous studies have been conducted since 2012. A comparison with 5-6 existing algorithms and approaches, such as Yolo (J. Whipp and A. Dong, "YOLO-based Deep Learning to Automated Bacterial Colony Counting," 2022 IEEE Eighth International Conference on Multimedia Big Data (BigMM), Naples, Italy, 2022, pp. 120-124, doi: 10.1109/BigMM55396.2022.00028), COVASIAM, etc., is vital.

5. The absence of an ethics committee protocol for data acquisition is a concern.

As it stands, the work cannot be recommended for publication.

Comments on the Quality of English Language

 Minor editing of English language required.

Reviewer 2 Report

Comments and Suggestions for Authors

In this work, an automatic colony counting system based on CNN-assisted automatic counting method was developed to count. But some aspects should be well descripted and answered.

1  With the increaseing aggregation of colonies, what is the recognition rate of the proposed U2-net model. Please state it.

2 In Figure 1B, 8 bacterial species plates can be devided into two classes due to their morphological features, and how about the difference about the counting accuacy and their influences ? It should be stated in the Discussion section.

3  Results comparsions of the proposed U2-Net and the metioned traditional segmentions should be made. Besdies, the accuracy for 8 bacterial countings should be tabled.

4 Many discriptions about the Imaging captures, Image preprocessing, Segmentation and others, They are the methodnology descriptions, and should be move the Section 2 Materials and methods.

5 In the conclusions, the results of the prediction accuracy should be listed out.

6 The supplienmentary materials should provided more about your experimental process. The CMOS and Len should be listed the brand and model.